# Novel Fabrication of 3-D Cell Laden Micro-Patterned Porous Structure on Cell Growth and Proliferation by Layered Manufacturing

**DOI:** 10.3390/bioengineering10091092

**Published:** 2023-09-18

**Authors:** Won-Shik Chu, Hyeongryool Park, Sangjun Moon

**Affiliations:** 1Department of Mechanical Convergence Engineering, Gyeongsang National University, Changwon 51391, Gyeongsangnam-do, Republic of Korea; wschu@gnu.ac.kr (W.-S.C.); parkhyeongryool@gmail.com (H.P.); 2Cyberneticsimagingsystems Co., Ltd., Changwon 51391, Gyeongsangnam-do, Republic of Korea

**Keywords:** cell-laden, hydrogel, scaffold, porous structure, layered manufacturing

## Abstract

This study focuses on developing and characterizing a novel 3-dimensional cell-laden micro-patterned porous structure from a mechanical engineering perspective. Tissue engineering holds great promise for repairing damaged organs but faces challenges related to cell viability, biocompatibility, and mechanical strength. This research aims to overcome these limitations by utilizing gelatin methacrylate hydrogel as a scaffold material and employing a photolithography technique for precise patterned fabrication. The mechanical properties of the structure are of particular interest in this study. We evaluate its ability to withstand external forces through compression tests, which provide insights into its strength and stability. Additionally, structural integrity is assessed over time to determine its performance in in vitro and potential in vivo environments. We investigate cell viability and proliferation within the micro-patterned porous structure to evaluate the biological aspects. MTT assays and immunofluorescence staining are employed to analyze the metabolic activity and distribution pattern of cells, respectively. These assessments help us understand the effectiveness of the structure in supporting cell growth and tissue regeneration. The findings of this research contribute to the field of tissue engineering and provide valuable insights for mechanical engineers working on developing scaffolds and structures for regenerative medicine. By addressing challenges related to cell viability, biocompatibility, and mechanical strength, we move closer to realizing clinically viable tissue engineering solutions. The novel micro-patterned porous structure holds promise for applications in artificial organ development and lays the foundation for future advancements in large soft tissue construction.

## 1. Introduction

Tissue engineering is one of the most promising methods to fabricate tissue in sufficient quantities to repair diseased or defective organs [1,2,3,4,5,6,7]. It serves as a motivating factor for tissue engineering to be successfully implemented in regenerative medicine and to develop and establish it as a clinically viable option. Regenerated microtissue suited for a specific organ’s form and functionality is primarily tested for in vitro viability [8,9]. Implantation of tissue ‘engineered’ in vitro by seeding cultured cells into a biomaterial scaffold, replacing or renewing functional tissue, could be achieved. Researchers tried fabricating a three-dimensional (3D) cell-laden matrix to develop functional replacement tissues [10,11,12,13]. However, few were successful in their in vitro development, and fewer were applied to test their effectiveness for in vivo conditions because of evident limitations [14]. Unpredictable cell viability, biocompatibility, and lack of mechanical strength are issues in tissue engineering [15]. Cell viability is one of the most important considerations among the intricate conditions to be fulfilled. Due to a lack of mass transport and limitations in nutrient exchange, cell necrosis occurs predominantly in large 3D scaffolds [16].

To solve this problem, previous research focused on fabricating a vascular network within the structure to achieve the desired fluidic exchange of materials, which is crucial for nutrient exchange and waste riddance. This is further highlighted when handling a larger tissue structure, often called for in artificial organ development, as cell necrosis impedes stable cell proliferation [17]. Biocompatibility is another consideration for the successful application of engineered tissue. Thermal, pH, and various environmental conditions are to be fulfilled throughout cell embedding and growth steps to ensure the desired rate of cell proliferation [18]. Hydrogels have been widely used as scaffolds for tissue growth because of their easy manipulation, biodegradability, and biocompatibility characteristics [19]. Although agarose’s structure holds advantages in its hydrophilicity and electrical neutrality, it is immunogenic and nondegradable. Thus, various classes and forms of hydrogels are employed for clinical studies.

And lastly, the mechanical properties of the scaffold structure Usually, porosity concomitantly results in sacrificing the entire engineered tissue’s structural integrity. The engineered tissue needs to withstand constant and often critical external forces. Hence, such properties are to be thoroughly evaluated in searching for suitable materials and developing a proper cell-laden structure.

Multi-layer photopatterning has been in the spotlight for engineering micro- and nano-structures with higher resolution of up to 50 µm and flexibility in design over direct inkjet printing or micro-extrusion and micro-valve [20]. Photo patterning, with its high resolution and flexibility in designing parameters, does not lack printing speed and repeatability. Thus, it was an optimum fabrication method for the mass production of repeated structures. Some of the research shows the effect of porous structures; however, the structures are not well defined [21,22,23,24]. Other studies show the controlled geometry of pore and piller; however, the materials are not hydrogel, which makes it hard to encaptulate the cells [25,26,27].

This study aims to develop a novel 3D cell-laden micro-patterned porous structure and assess its cell viability, biocompatibility, and mechanical strength. A cell-laden structure with HUVEC and W-20-17 cell lines was used to evaluate the structure’s cell viability. Additionally, the assessment of its structural integrity is determined through mechanical tests. The novel photolithography technique achieved the porous structure, which carries comparable advantages over other scaffold designs because it provides uniform spacing for the successful exchange of biomaterials. The layer-by-layer photolithography technique enables systematic design and manual manipulation of the cell-laden structure. Furthermore, microchannels designed and built based on subsequent photolithography for material exchange may be potentially helpful to supply oxygen and nutrients to the fabricated tissue constructs, which will lay the foundation for developing microfluidic-directed assembly. Thus, the fabrication of porous structures by photomask was realized, which serves as a platform for large soft tissue construction.

Novel layered manufacturing was employed to fabricate controllable porous structures with accurate and consistent layers. Multi-layer structures with varying widths were evaluated based on their medium diffusion by distance from the culture medium. After 3 and 7 days of cell encapsulation, the cell behavior was tested by microscopic analysis.

## 2. Materials and Methods

### 2.1. Materials

Gelatin (type A, 275–325 bloom from porcine skin) and methacrylic anhydride (MA) were purchased from Sigma-Aldrich for the synthesis of GelMA. Glass slides and coverslips were purchased from Fisher Scientific. Fetal bovine serum (FBS), Trypsin–EDTA, l-glutamine, antibiotic–antimycotic, phosphate-buffered saline (PBS), and Dulbecco’s modified Eagle’s medium (DMEM) were all purchased from Invitrogen™ (Eugene, OR, USA). MA has acute toxicity when ingested or inhaled and can cause skin and eye damage upon exposure. Therefore, the material was handled with great care, following the material safety data sheet.

### 2.2. GelMA Synthesis

GelMA was synthesized as previously described [28,29]. Briefly, 10% (*w*/*v*) gelatin powder was mixed with Dulbecco’s phosphate-buffered saline (DPBS, Gibco) and stirred until completely dissolved at 50 °C. In this research, a two-vol% MA was used in the synthesis reaction to achieve a low degree of methacrylation, as previously demonstrated [28]. MA was added at a rate of 0.5 mL/min at 50 °C under stirring conditions and allowed to react for 2 h. Afterward, the solution was diluted 5X with DPBS to halt the reaction. The mixture was dialyzed against distilled water using 12–14 kDa cutoff dialysis tubing at 40 °C for one week to remove salts and unreacted methacrylic acid. Finally, the solution was lyophilized for three days to generate a white, porous foam and stored at −80 °C for further experiments. Micro-patterned porous structures were fabricated using simplified 3D-printing technology combined with photolithography technology. Instead of using a selective curing method by laser for 3D structure fabrication, this research employed repeated photolithography using a custom device with a photomask containing intended micropatterns, as shown in Figure 1. The stage, equipped with up-and-down motion, was set up with a micro-patterned photomask. A spacer (colored orange in Figure 1) was added to each layer to limit the thickness of the structure. After placing the spacer, the hydrogel was poured into the spacer, and a mask was placed on the setup. The hydrogel was then cured using UV (Ultra Violet) light, and steps 2 to 4 in Figure 1 were repeated until the desired number of layers in the structure was achieved (a single or multi-layered structure can be achieved). Figure 2 provides a schematic representation of the micro-patterned porous structures in this research with different porosities.

### 2.3. Design of Micro-Patterned Porous Structures

Three different specimens were prepared to observe the effect of the channel structure in the hydrogel. The micro-pattern structure was created by stacking square pillar unit structures in a checkered orientation. The gaps between the pillars provide porosity between neighboring layers. Since the diameter of the mouse aorta and artery ranges from 150 to 535 µm, we designed the gap of the structure to be 0, 250, and 500 µm [30] to supply nutrients to encapsulated cells. The structure size is determined by multiplying each unit pillar’s length, width, height, and the number of structures. Therefore, the structure’s size represents the volume of the entire structure. Both the structure size and cell count remained constant throughout to enable quantitative comparisons of cell activities in samples with different porosities. The volume was limited to 13.69 mm^3^, and the cell density was 4930 cells/m^3^ for each structure. Consequently, the width and height of the specimens were designed differently while maintaining a constant number of cells. The thickness of each layer was 500 µm, and the total thickness was controlled at 1000 µm.

This is directly linked to the amount of material (GelMA) required to build the micro-porous structure. The cell count, another factor kept constant with the structure size, represents the number of cells initially embedded in the structure and was used to test cell viability under identical conditions. Table 1 provides information related to the structure and cell number of this research.

### 2.4. Hydrogel Preparation

To prepare a photoinitiation (PI) solution, we first mixed 0.5% (*w*/*v*) of 2-hydroxy-1(4-hydroxymethyl) phenyl)-2-methyl-1-propanone (Irgacure 2959, CIBA Chemicals, Toms River, NJ, USA) with DPBS at 80 °C until it was fully dissolved. Next, we added 5% (*w*/*v*) GelMA to the PI solution and stirred it until it was completely dissolved. To remove any bubbles, the solution was placed in an oven at 60 °C for 5 min. Previous research has shown that, for proper structural handling in subsequent processes, GelMA concentration should be at least 5% (*w*/*v*).

### 2.5. Fabrication of Micro-Patterned Porous Structures

Each specimen was fabricated using the novel layered manufacturing techniques mentioned previously. To create various patterns, photomasks with no pattern, 250 µm, and 500 µm channels in width, respectively, were designed using AutoCAD software and printed on transparencies with a resolution of 10,000 dpi (CAD/Art Services, Bandon, OR, USA). These photomasks were placed on the microstage for further processing. The micro-stage was thoroughly cleaned and tested for debris before constructing the base layer of the porous structure. It was also prepared to align the photomask equidistant from the base layer of the stage and to place the spacer to hold and position the gel material. The photomask selectively cured the material to build a single layer of pillars. The pre-polymer, guided out by the photomask but left uncured, was removed, and these steps were repeated to construct the targeted multi-layer porous structure.

### 2.6. Mechanical Tests

For the application in in vitro/in vivo environments, it is important for the micro-patterned porous structure to maintain its morphology during the experimental period. Five specimens were marked at each time point and fabricated as described in Section 2.4. They were then incubated in a buffer solution for seven days to test their comparative compressive modulus. The test was conducted using an Instron 4465 mechanical tester at room temperature, with a 5 kN load cell, following the guidelines of ASTM (other conditions were followed in previously conducted research) [28]. The dimensions of the specimens were 3.75 mm^2^ × 1 mm, 4.5 mm^2^ × 1 mm, and 5.4 mm^2^ × 1 mm for specimens with no pattern, 250 μm pattern, and 500 μm pattern samples, respectively. A schematic diagram of the compressive stress test is shown in Figure 3.

### 2.7. Cell Culture and Cell Seeding

To evaluate cell proliferation and differentiation on micro-patterned porous structures, human umbilical vein endothelial cells (HUVECs) and W-20-17 mouse bone marrow stromal cells were seeded into the micro-patterned porous structure for 1 and 7 days, respectively. The cell counts are provided in Table 1. HUVECs were obtained from the laboratory of the late Dr. J. Folkman (Children’s Hospital, Boston, MA, USA) and cultured in EBM-2, which contains supplements from an EGM-2 kit and 10% FBS. W-20-17 cells were purchased from ATCC (American Type Culture Collection, Manassas, VA, USA) and cultured in DMEM media with 10% FBS, a 1% antibiotic/antimycotic mixture, 5 mL of an L-glutamine solution (200 mM), and sodium pyruvate. Both cell types were cultured in an incubator maintained at 5% CO_2_ and 37 °C.

### 2.8. Cell Activity

#### 2.8.1. MTT Assay

A total of 3-(4,5-dimethylthiazol-2-yl)-2,5-diphenyl tetrasodium bromide (MTT) was used to quantify the cell’s metabolically active status. On days 1 and 7, each sample was digested with a 1 mg/mL collagenase-A solution for 20 min at 37 °C. Cells were retrieved and transferred into a 1.5-mL centrifuge tube. Then, 1 mL of serum-free growth medium that contains 100 µL 5 mg/mL MTT was added (1/10; final concentration was 0.5 mg/mL) at each time point (days 1 and 7). Each tube was then placed to incubate at 37 °C for 4 h. During incubation, MTT was taken up by active cells and reduced in the mitochondria to insoluble purple formazan granules. The formazan crystals formed were solubilized by adding 0.5 mL of dimethyl sulfoxide for 30 min under shaking. The optical density of formazan in the solution was then read using a microplate reader at 490 nm (BIO-RAD Model 680, Bio-Rad Laboratories, Inc., Hercules, CA, USA).

#### 2.8.2. Alkaline Phosphatase (ALP) Activity

To assess cell proliferation, the concentration of double-stranded DNA (dsDNA) was quantified using a fluorometric assay. W-20-17 cells were seeded onto the fabricated micro-patterned porous structures for 1 and 7 days. At each time point, cells were lysed in a 0.2% Triton X-100 solution, followed by three freeze (−80 °C) and thaw (37 °C) cycles. To measure ALP activity with a p-nitrophenyl phosphate (p-NPP) substrate, as described elsewhere, cell lysates were transferred and placed in a 96-well plate [31]. After a 30-min incubation at 37 °C, absorbance values were measured at 405 nm using a microplate reader (BIO-RAD 680, USA). The total amount of protein in the cell lysates was determined using a micro-BCA Protein Assay Kit (Thermo Scientific, Waltham, MA, USA). Finally, the total protein concentration was expressed as units per gram of protein by normalizing ALP activity.

#### 2.8.3. Immunofluorescence Staining

Protein components and the distribution pattern of HUVEC-derived extracellular matrix (ECM) in the micro-patterned porous structure were verified with immunofluorescence staining. The immunofluorescence staining was used for testing the cell viability of HUVEC cells, indicating the rate of cell proliferation under adverse conditions. The results compare the live and dead cells within the specific region with a live cell indication by the activated region of cells lightening up under UV rays.

## 3. Results and Discussion

### 3.1. Mechanical Characterization

Tensile tests were conducted on the substrates to assess their suitability for use in various in vitro and in vivo environments. Figure 4 summarizes the compressive modulus of these specimens. The elastic modulus (*E*) is characterized as the slope of stress (*σ*) relative to elongation (*ε*), highlighting how much stress a particular material can withstand before undergoing plastic deformation. *E* can be used to characterize micro-porous structures optimized for use in artificial organs. Comparing the deformation behavior of specimens with no lattice pattern to those with 500 μm and 250 μm spaced lattice patterns, we tested their ability to withstand compression. It is evident that as the distance between micro-porous spaces increases, the structure becomes better at withstanding compression, as deduced from the 500 μm patterned modulus of specimens compared to the 250 μm patterned ones. Furthermore, after incubation in a buffer solution for a week, the specimens exhibited a general decrease in compressive modulus, which is expected due to the weakening of the material’s chemical bonds over time in an in vitro environment.

However, the compressive modulus of the porous structure was similar to that of the specimen with no pattern. The structure with no porosity performed similarly in the compression test to the porous structure. On day 1 of the experiment, the specimen with no porosity exhibited a compressive modulus ranging between 0.38 and 0.40 kPa, falling within the close range of specimens designed with a 500 μm porous pattern. The specimens with 250 μm patterns showed a slightly lower compressive modulus compared to the previously mentioned specimens. On day 7 of the experiment, both specimens with 500 μm and 250 μm patterns exhibited an expected decrease in their compressive modulus, similar to the control specimen with no pattern. Therefore, it can be inferred that porosity has no significant impact on the mechanical properties of the material, and structures with higher porosity performed better in compression evaluations.

When comparing the results over time, specimens embedded with porosity appear to be a viable option for applications, and their mechanical performance can be on par with the previously tested specimens with no pattern. Furthermore, it can be concluded that the porous, structured hydrogel exhibits a sufficient level of mechanical stability for medical applications. To support tissue and organ regeneration, it should maintain prolonged stiffness and elastic modulus until it grows to replace the original tissue. Additionally, to facilitate the mass transport of nutrients and other necessary materials for cell proliferation and, ultimately, tissue growth, the structure must withstand certain pressures caused by fluid flow. The micro-patterned porous structure has been shown to have comparable mechanical properties suitable for use in such conditions.

### 3.2. Viability

#### HUVEC

Cell viability tests were performed on specimens to assess their long-term effects. The MTT assay was conducted on specimens with 500 μm and 250 μm patterns, as well as those with no pattern. The results, presented in Figure 5, show the absorbance rate measured by the amount of formazan granules present.

On Day 1, as expected, substrates with 500 μm and 250 μm spaced patterns exhibited a higher absorbance rate than the control. The micro-porous channels may have acted as pathways for the staining agent to reach the cultures, resulting in a 42% higher absorbance rate. When comparing the Day 1 results with the final result on Day 7, it is clear that the absorbance rate increased over time.

The comparative results of the 500 μm and 250 μm channel-patterned structures are visualized through immunofluorescence staining (Figure 6 and Figure 7). These tests were performed on samples on both Day 1 and Day 7 of the experiment, revealing the distribution pattern of HUVEC cell lines. The vision of the first and second layers of the seeded microporous substrate was characterized based on active cell interactions within each layer. From the experimental results of Day 1 and Day 7, there is an increase in cell count and density for both 500 μm and 250 μm patterned specimens. The specimen with the 500 μm pattern shows a faster proliferation rate compared to the 250 μm pattern.

DAPI and DsRed staining were additionally conducted to confirm the increase in fixed cell proliferation. The stained sample from Day 7 shows steady growth in active cells seeded on hydrogel layers (Figure 8). Figure 9 displays the live/dead staining results from Days 1 to 7. All the structures form cellular networks after Day 7, with even the no-pattern specimens showing a few dead cells within the network structure.

### 3.3. W-20-17

The MTT assay was performed on W-20-17 cells to investigate the differentiation pattern of another cell line used for artificial organ development (Figure 10). The absorbance of formazan, indicating the comparative number of active cells within a volume of culture with an equal cell density, shows a higher number of active cells on the micro-patterned layer with a 500 μm width on Day 1 compared to the specimen with a 250 μm pattern. This contrasted notably with the control with no pattern, where absorbance remained within the range of 0.4 and 0.55. In contrast, the no-pattern specimens showed only a slight increase in absorbance on Day 7.

An ALP study was conducted on W-20-17 to verify the longevity of the cell-laden patterns with varying widths (Figure 11). Comparative results from Day 1 and Day 7 demonstrate that cell proliferation was higher on constructs with patterns compared to the bulk hydrogel. Figure 12 illustrates the characteristics of embedded W-20-17 cells in micropatterned structures, with proliferation occurring at a slower rate compared to HUVEC. However, all W-20-17 specimens show dead cells within the pattern, unlike HUVEC.

## 4. Conclusions

In this study, we designed and described a micro-porous structure with high mechanical strength and characterized the differentiation patterns of HUVEC and the W20-17 cell line. Micro-porosity was achieved through manual manipulation of matrix layering patterns via the design of a photomask.

The results of this research provide a viable option for artificial organ studies. The photopatterning and layering of micro-layers of hydrogel substrate seeded with cells to develop a novel micro-porous structure open up new possibilities for regenerative medicine. The proliferation patterns of HUVEC and MSC, along with the w20-17 cell line on the developed hydrogel layers, indicate that microporosity significantly affects cell viability, especially over time. Microporosity serves as the conduit for substrate and nutrient exchange, making it a promising structure for creating larger engineered tissues.

Current limitations in tissue engineering are often linked to cells undergoing apoptosis due to insufficient substrate exchange. However, the developed microporous structure offers an advantage in terms of maintaining a higher number of viable cells over an extended period without sacrificing mechanical properties compared to previously researched unpatterned hydrogels. This structure, with its porous design, demonstrates mechanical stability suitable for various applications.

It lays the groundwork for the development of large soft tissue constructs, overcoming previous limitations related to the 2-dimensional design of cell layers. Microporous structures can be easily modified with different pattern designs, and this research showcases the potential for manipulating such structures through the technology presented herein.

## Figures and Tables

**Figure 1 bioengineering-10-01092-f001:**
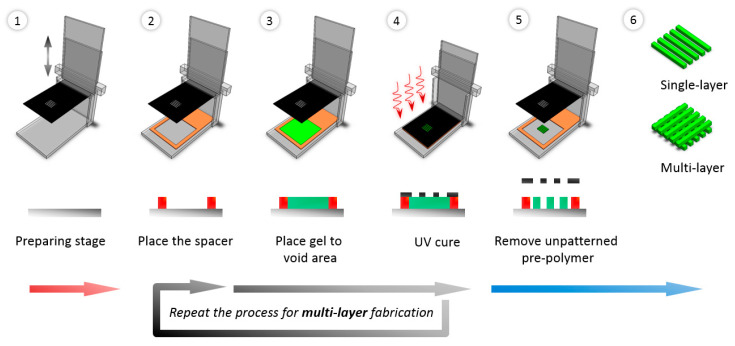
A multi-layer fabrication process for 3-dimensional micro-patterned structures using layered manufacturing The mask (black in the figure) was used to generate patterns on a micro-scale, and the device provides a repeatable process with an up-and-down motion of the mask.

**Figure 2 bioengineering-10-01092-f002:**
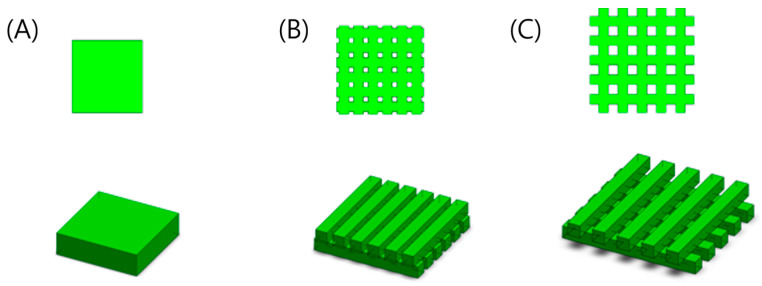
Schematic of micro-patterned porous structures using the layered manufacturing method. (**A**) is the control specimen that has no channel, and (**B**,**C**) have 33.3% and 53.7% porosity, respectively.

**Figure 3 bioengineering-10-01092-f003:**
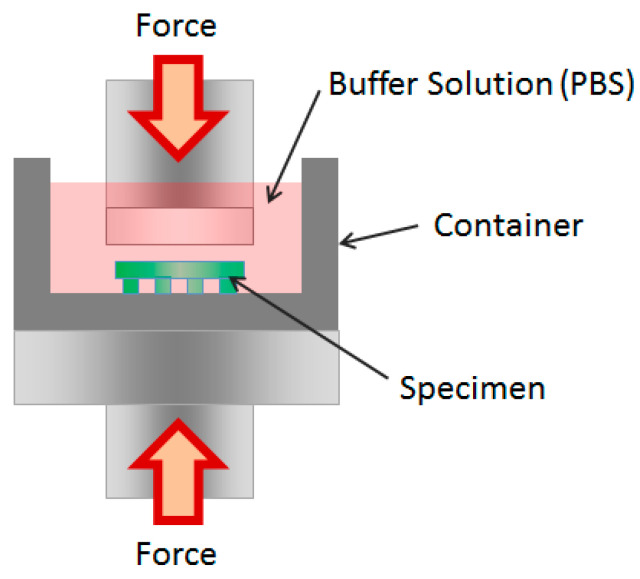
Schematic of the compressive modulus test setup.

**Figure 4 bioengineering-10-01092-f004:**
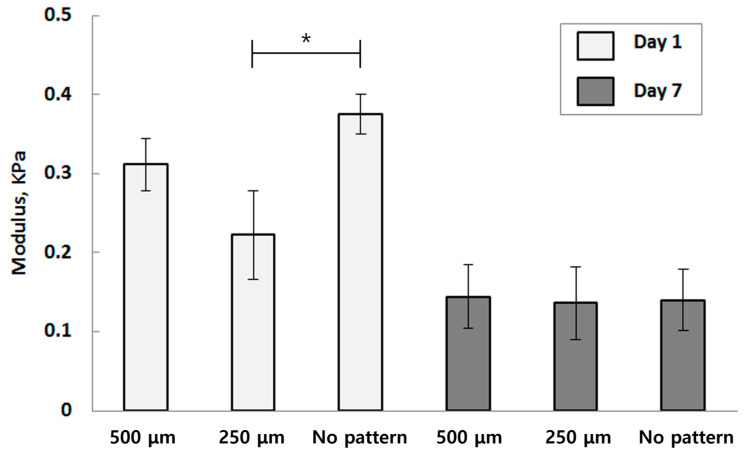
The compressive modulus of each structure with different time points (Each value represents the mean ± SD (*n* = 5; * donates a significant difference (*p* < 0.05)).

**Figure 5 bioengineering-10-01092-f005:**
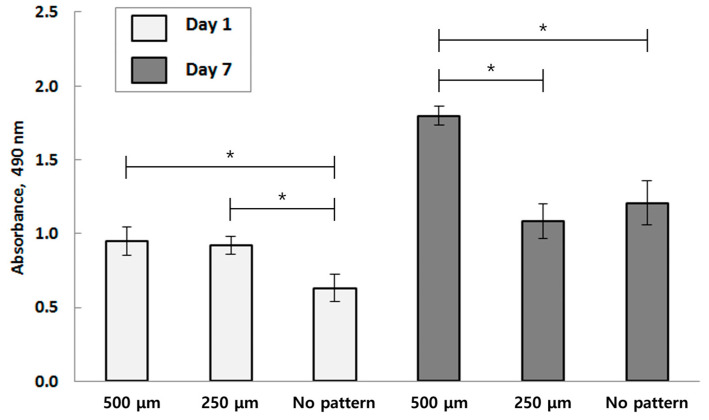
Quantitative measurement of the amount of the formazan present in the HUVECs seeded in each pattern by MTT assay (Each value represents the mean ± SD (*n* = 5, * donates a significant difference (*p* < 0.05)).

**Figure 6 bioengineering-10-01092-f006:**
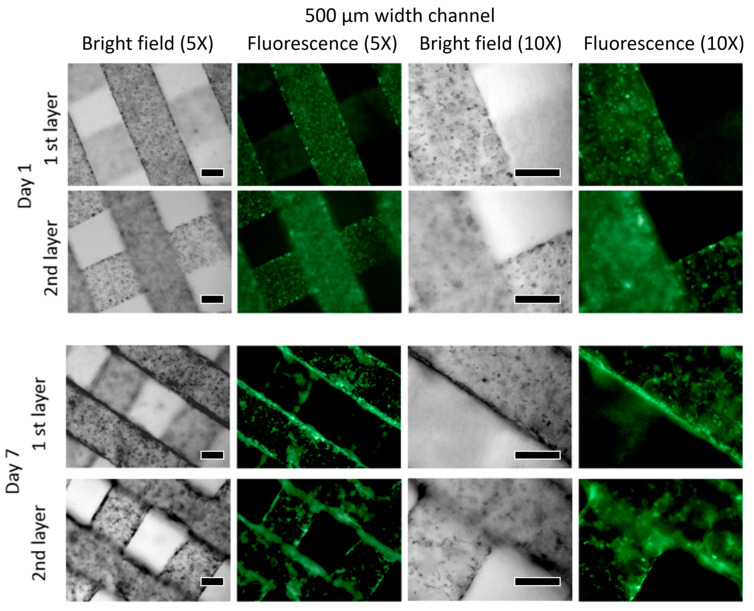
A total of 500 μm channel patterned structure is visualized by immunofluorescence staining (the scale bar indicates 250 μm).

**Figure 7 bioengineering-10-01092-f007:**
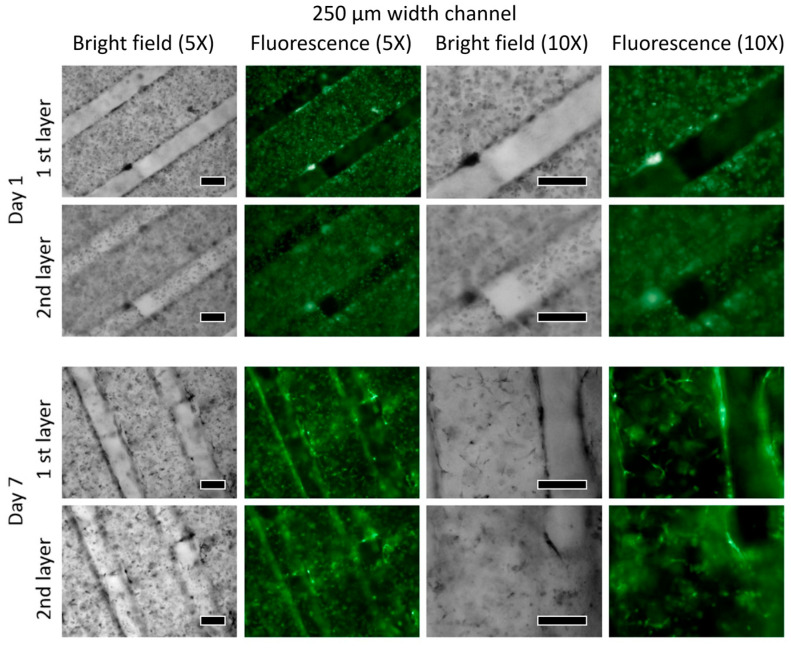
A total of 250 μm channel patterned structure is visualized by immunofluorescence staining (the scale bar indicates 250 μm).

**Figure 8 bioengineering-10-01092-f008:**
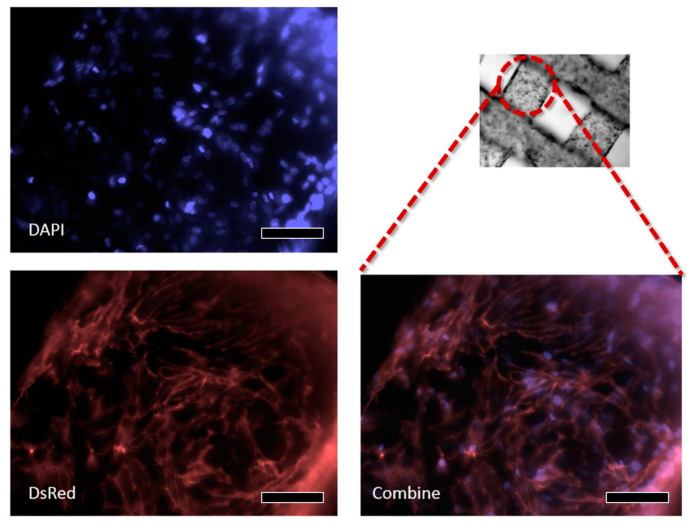
A result DAPI and DsRed staining after Day 7 (the scale bar indicates 100 μm).

**Figure 9 bioengineering-10-01092-f009:**
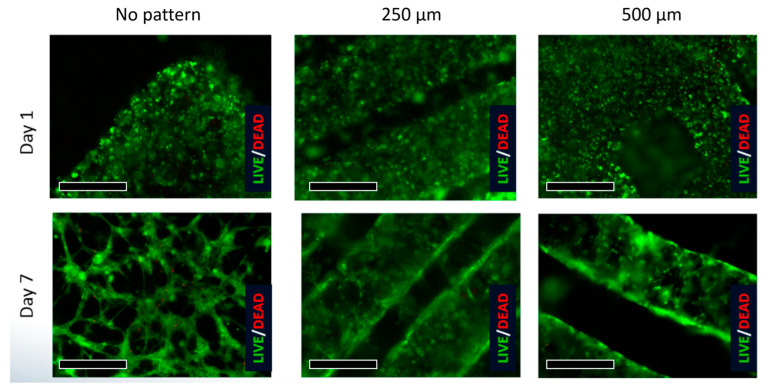
Characterization of embedded HUVEC cells’ behavior in micropatterned structures stained with the IVE/DEAD assay (the scale bar indicates 500 μm).

**Figure 10 bioengineering-10-01092-f010:**
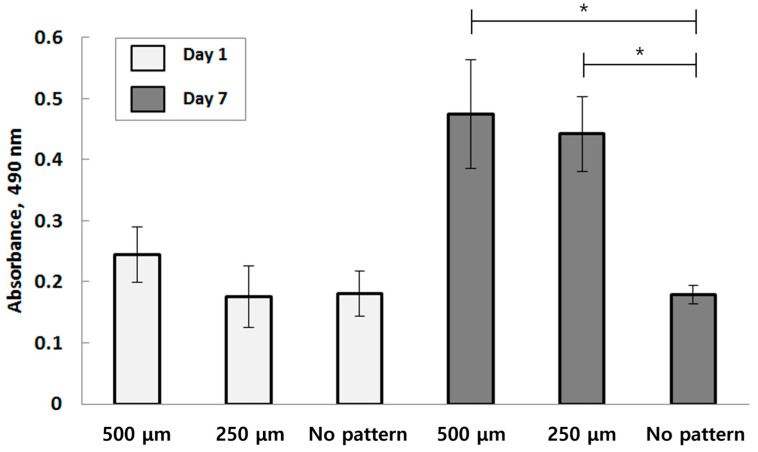
Quantitative measurement of the amount of the formazan present in the W-20-17s seeded each pattern by MTT assay (Each value represents the mean ± SD (*n* = 5; * donates a significant difference (*p* < 0.05)).

**Figure 11 bioengineering-10-01092-f011:**
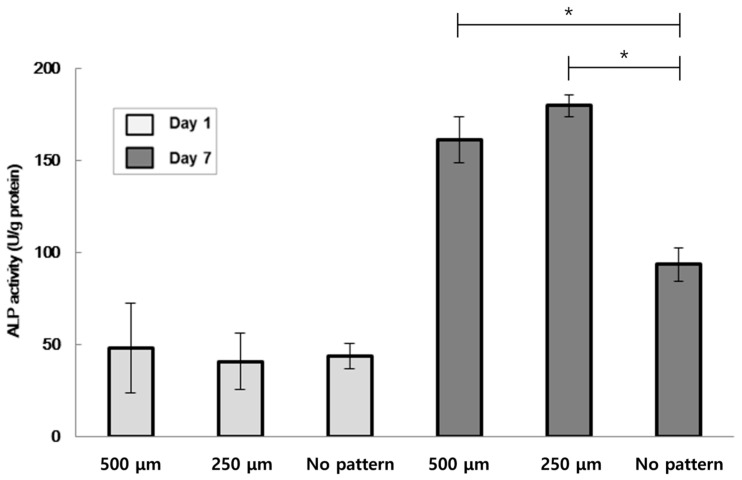
ALP activity of the W-20-17 cell-laden patterns (Each value represents the mean ± SD (*n* = 5, * donates significant difference (*p* < 0.05)).

**Figure 12 bioengineering-10-01092-f012:**
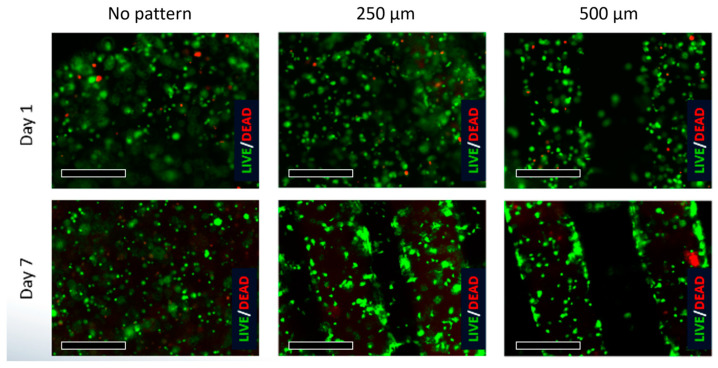
Characterization of embedded W-20-17s cells behavior in micropatterned structures stained with LIVE/DEAD assay (the scale bar indicates 500 μm).

**Table 1 bioengineering-10-01092-t001:** Structure size and its characteristics.

	Structure Size (Length × Width × Height × No. of Structures)	Channel Size (mm) Stand-Off Distance (SD)	Porosity (%)	Cell Number	Surface Area (mm^2^)
A	3.7 mm × 3.7 mm × 1 mm × 1	0.00	00.0%	67,500	42.18
B	4.5 mm × 0.5 mm × 0.5 mm × 12	0.25	33.3%	67,500	96.00
C	5.4 mm × 0.5 mm × 0.5 mm × 10	0.50	53.7%	67,500	100.00

## Data Availability

All relevant data generated or analyzed during this study are included in this article. Further enquiries can be directed to the corresponding author.

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
