# Peer review of "Novel Fabrication of 3-D Cell Laden Micro-Patterned Porous Structure on Cell Growth and Proliferation by Layered Manufacturing"

_bioengineering, 2023, doi:10.3390/bioengineering10091092_

Round 1

Reviewer 1 Report

The article entitled „Novel Fabrication of 3-D Cell Laden Micro-patterned Porous Structure on Cell Growth and Proliferation by Layered Manufacturing” by Chuet al. is an interesting paper. It is particularly valuable to think about tissue engineering for possiblereparation of affected organs. The methods used are suitable for the intended purpose, and the results are notable.

Is there an assumption why the mechanical characteristics became the same for all the samples on Fig 4 after 7 days regardles of the pore size?

More detailed explanation and evaluation of the test results would be benefitial. 

Minor revision is suggested for the paper.

It's alright.

Author Response

Answers to the reviewer’s comments

On behalf of my co-authors, I am re-submitting the enclosed material after revision for possible publication in your journal. We sincerely thank the reviewers for their careful reading of the manuscript and valuable comments. We have made revisions according to the reviewers’ helpful comments and suggestions, as described below. The revised portions of the manuscript are highlighted in blue.

The article entitled „Novel Fabrication of 3-D Cell Laden Micro-patterned Porous Structure on Cell Growth and Proliferation by Layered Manufacturing” by Chu et al. is an interesting paper. It is particularly valuable to think about tissue engineering for possible reparation of affected organs. The methods used are suitable for the intended purpose, and the results are notable.

Is there an assumption why the mechanical characteristics became the same for all the samples on Fig 4 after 7 days, regardless of the pore size? More detailed explanation and evaluation of the test results would be benefitial. Minor revision is suggested for the paper.

Ans.: On day 1, we attributed the variation in mechanical characteristics among the specimens to geometric factors. In contrast, for the specimens on day 7, we speculated that the similarity in mechanical properties might be a result of cell proliferation and hydrogel degradation.

Reviewer 2 Report

In this study authors prepared micro patterned porous structure for cell growth and proliferation. Following comments should be addressed.

1. Abstract should start from rationale of the study.

2. There are many spelling and grammatical mistakes in the manuscript. Please revise it thoroughly.

3. What is new and novel in this study beyond the already reported literature?

4.  Please cite some latest papers of the respective journal.

5. Results are not supported with relevant literature. Please do the needful.

6. Please highlight the novelty of the study.

English should be improved.

Author Response

Answers to the reviewer’s comments

On behalf of my co-authors, I am re-submitting the enclosed material after revision for possible publication in your journal. We sincerely thank the reviewers for their careful reading of the manuscript and valuable comments. We have made revisions according to the reviewers’ helpful comments and suggestions, as described below. The revised portions of the manuscript are highlighted in blue.

In this study authors prepared micro patterned porous structure for cell growth and proliferation. Following comments should be addressed.

  1. Abstract should start from rationale of the study.

Ans.: The abstract was modified as follows

(Line : 10-15).

Tissue engineering holds great promise for repairing damaged organs but faces challenges related to cell viability, biocompatibility, and mechanical strength. This Study focused on the development and characterization of a novel three-dimensional cell-laden micro-patterned porous structure from a mechanical engineering perspective to overcome these limitations by utilizing gelatin methacrylate hydrogel as a scaffold material and employing photolithography techniques for precise patterned fabrication.

  1. There are many spelling and grammatical mistakes in the manuscript. Please revise it thoroughly.

Ans.: The entire manuscript was revised following the reviewer’s comment.

  1. What is new and novel in this study beyond the already reported literature?

Ans.: The purpose of this study is to fabricate micro-patterned 3D structures using simple layered manufacturing and analyze these micro-patterned 3D structures using the device. The following explanation has been added to the manuscript.

(Line: 108-112)

Micro-patterned porous structures were fabricated using simplified 3D-printing technology combined with photolithography technology. Instead of using a selective curing method by laser for 3D structure fabrication, this research employed repeated photolithography using a custom device with a photomask containing intended micro patterns, as shown in Figure 1.

  1. Please cite some latest papers of the respective journal.

Ans.: The latest research studies were cited in the manuscript.

  1. Results are not supported with relevant literature. Please do the needful.

Ans.: Relevant works of literature were cited in the manuscript

  1. Please highlight the novelty of the study.

Ans.: This study focuses on developing and characterizing a novel 3-dimensional cell-laden micro-patterned porous structure from a mechanical engineering perspective. This research aims to overcome these limitations by utilizing gelatin methacrylate hydrogel as a scaffold material and employing a photolithography technique for precise patterned fabrication. The mechanical properties of the structure are of particular interest in this study. We evaluate its ability to withstand external forces through compression tests, which provide insights into its strength and stability. Additionally, the structural integrity is assessed over time to determine its performance in in vitro and potential in vivo environments. By addressing challenges related to cell viability, biocompatibility, and mechanical strength, we move closer to realizing clinically viable tissue engineering solutions. The novel micro-patterned porous structure holds promise for applications in artificial organ development and lays a foundation for future advancements in large soft tissue construction.

Reviewer 3 Report

The paper entitled “Novel Fabrication of 3-D Cell Laden Micro-patterned Porous Structure on Cell Growth and Proliferation by Layered Manufacturing is an interesting area of study. The paper is written well but there are a few suggestions and limitations of the study that should be incorporated before it gets published.

1.       The section is well-organized and clearly delineated into subsections, which aids readability. Each subsection provides a detailed description of the methods used, which is crucial for the reproducibility and understanding of the study.

2.       The materials and reagents used are well-documented, which is essential for transparency in scientific research. Consider providing the sources or suppliers for the specific equipment used in the study (e.g., Instron 4465 mechanical tester) for clarity.

3.       The GelMA synthesis process is well-described, but it might be helpful to include the rationale for using specific conditions and concentrations. Mention safety precautions if any hazardous chemicals or procedures are involved.

4.       Design of Micro-Patterned Porous Structures could benefit from more detailed information on the specific dimensions and characteristics of the micro-patterned structures. Provide specific numerical values. Explain the reasoning behind choosing these specific dimensions and structures.

5.       In Hydrogel Preparation Include the rationale for using 2-hydroxy-1(4-hydroxymethyl) phenyl)-2-methyl-1-propanone (Irgacure 2959) as the photo initiation solution. Why was this particular compound chosen?

6.       Fabrication of Micro-Patterned Porous Structures Provides more details on the photolithography process, including exposure times, wavelengths, and intensities used during UV curing. Explain how the spacing between layers is controlled during the fabrication process.

7.       The mechanical testing section is informative, but it could benefit from additional context. Why were these specific tests chosen, and what standards or guidelines were followed? Clarify the units used in reporting the compressive modulus (e.g., KPa or MPa).

8.       Mention the seeding density for both HUVEC and W-20-17 cells. This information is crucial for understanding the cell culture conditions.

9.       Provide more context for the choice of cell activity assays (MTT, ALP). Explain why these specific assays were chosen to evaluate cell viability and proliferation.

1.   Consider providing the full names of abbreviations (e.g., MTT) for clarity.

1.   Ensure that figures and data are properly labeled, the statistical significance will be added and axis labels are clear and informative. Mention the scale or magnification for microscopy images to give readers a sense of size.

Minor poslishing required 

Author Response

Answers to the reviewer’s comments

On behalf of my co-authors, I am re-submitting the enclosed material after revision for possible publication in your journal. We sincerely thank the reviewers for their careful reading of the manuscript and valuable comments. We have made revisions according to the reviewers’ helpful comments and suggestions, as described below. The revised portions of the manuscript are highlighted in blue.

The paper entitled “Novel Fabrication of 3-D Cell Laden Micro-patterned Porous Structure on Cell Growth and Proliferation by Layered Manufacturing is an interesting area of study. The paper is written well but there are a few suggestions and limitations of the study that should be incorporated before it gets published.

  1. The section is well-organized and clearly delineated into subsections, which aids readability. Each subsection provides a detailed description of the methods used, which is crucial for the reproducibility and understanding of the study.
  2. The materials and reagents used are well-documented, which is essential for transparency in scientific research. Consider providing the sources or suppliers for the specific equipment used in the study (e.g., Instron 4465 mechanical tester) for clarity.
  3. The GelMA synthesis process is well-described, but it might be helpful to include the rationale for using specific conditions and concentrations. Mention safety precautions if any hazardous chemicals or procedures are involved.

Ans.: The paragraph related to the comment was modified, and an explanation was added.

(Line 95-97)

MA has acute toxicity when ingested or inhaled and can cause skin and eye damage upon exposure. Therefore, the material was handled with great care, following the material safety data sheet.

(Line: 150-156)

To prepare a photoinitiation (PI) solution, we first mixed 0.5% (w/v) of 2-hydroxy-1(4-hydroxymethyl) phenyl)-2-methyl-1-propanone (Irgacure 2959, CIBA Chemicals) with DPBS at 80 °C until it was fully dissolved. Next, we added 5% (w/v) GelMA to the PI solution and stirred it until it was completely dissolved. To remove any bubbles, the solution was placed in an oven at 60 °C for 5 minutes. Previous research has shown that, for proper structural handling in subsequent processes, GelMA concentration should be at least 5% (w/v).

  1. Design of Micro-Patterned Porous Structures could benefit from more detailed information on the specific dimensions and characteristics of the micro-patterned structures. Provide specific numerical values. Explain the reasoning behind choosing these specific dimensions and structures.

Ans.: The paragraph related to the comment was modified, and an explanation was added.

(Line: 133-141)

The gaps between the pillars provide porosity between neighboring layers. Since the diameter of the mouse aorta and artery ranges from 150 to 535 µm, we designed the gap of the structure to be 0, 250, and 500 µm [23] to supply nutrients to encapsulated cells. The structure size is determined by multiplying each unit pillar's length, width, height, and the number of each structure. Therefore, the structure size represents the volume of the entire structure. Both the structure size and cell count remained constant throughout to enable quantitative comparisons of cell activities in samples with different porosity. The volume was limited to 13.69 mm³, and the cell density was 4,930 cells/m³ for each structure.

  1. In Hydrogel Preparation Include the rationale for using 2-hydroxy-1(4-hydroxymethyl) phenyl)-2-methyl-1-propanone (Irgacure 2959) as the photo initiation solution. Why was this particular compound chosen?

Ans.: In this research, we used GelMA, which is a well-known material for cell-laden structures. The material is not the main issue; instead, it's the geometrical aspects. Therefore, most of the protocols for material preparation were followed as previously mentioned in the literature.

  1. Fabrication of Micro-Patterned Porous Structures Provides more details on the photolithography process, including exposure times, wavelengths, and intensities used during UV curing. Explain how the spacing between layers is controlled during the fabrication process.

Ans.: The below explanation was added to the manuscript.

(Line: 113-114)

A spacer (colored orange in Figure 1) was added to each layer to limit the thickness of the structure.

  1. The mechanical testing section is informative, but it could benefit from additional context. Why were these specific tests chosen, and what standards or guidelines were followed? Clarify the units used in reporting the compressive modulus (e.g., KPa or MPa).

Ans.: Since the mechanical test for the hydrogel differs from that of ordinary polymers, we followed the same ASTM guidelines, and other conditions that were not mentioned in the manuscript were previously employed in conducted research (ref. 28). The compressive moduli were expressed in kPa. The sentence has been modified as follows to provide further clarification.

(Line: 175-177)

The test was conducted using an Instron 4465 mechanical tester at room temperature, with a 5 kN load cell, following the guidelines of ASTM (other conditions were followed in previously conducted research) [28].

  1. Mention the seeding density for both HUVEC and W-20-17 cells. This information is crucial for understanding the cell culture conditions.

Ans.: The information is added as in Ans 4.

  1. Provide more context for the choice of cell activity assays (MTT, ALP). Explain why these specific assays were chosen to evaluate cell viability and proliferation.

Ans.: MTT Cell Growth Assay is a colorimetric assay that can be used for either proliferation or complement-mediated cytotoxicity assays. An MTT assay is a colorimetric assay that detects the color change from yellow of the tetrazolium dye to purple due to the formation of formazan in the presence of viable cells with active metabolism. ALP is an important marker for determining osteoblast phenotype. In this research, the concentration of double-stranded DNA (dsDNA) was quantified using a fluorometric assay to assess cell proliferation.

  1. Consider providing the full names of abbreviations (e.g., MTT) for clarity.

Ans.: The full name of MTT was added in the manuscript.

(Line: 196)

3-(4,5-dimethylthiazol-2-yl)-2,5-diphenyl tetrasodium bromide (MTT) was used to quantify the metabolic activity of the cells.

  1. Ensure that figures and data are properly labeled, the statistical significance will be added and axis labels are clear and informative. Mention the scale or magnification for microscopy images to give readers a sense of size.

Ans.: All the graphs and figures were modified as per the reviewer's comments.

Round 2

Reviewer 2 Report

Authors have addressed the comments adequately.

Reviewer 3 Report

Accepted in current form